# FedDecay: Adapting to Data Heterogeneity in Federated Learning With Gradient Decay

## Abstract

Federated learning is a powerful technique for collaboratively training a centralized model on distributed local data sources, preserving data privacy by aggregating model information without disclosing the local training data. However, the inherent diversity in local data sets challenges the performance of traditional single-model-based techniques, especially when data is not identically distributed across sources. Personalized models can mitigate this challenge but often have additional memory and computation costs. In this work, we introduce FedDecay, a novel approach that enhances single-model-based federated learning by incorporating gradient decay into local updates within each training round. FedDecay adapts the gradient during training by introducing a tunable hyper-parameter, striking a balance between initial model success and fine-tuning potential. We provide both theoretical insights and empirical evidence of FedDecay's efficacy across diverse domains, including vision, text, and graph data. Our extensive experiments demonstrate that FedDecay outperforms other single-model-based methods regarding generalization performance for new and existing users. This work highlights the potential of tailored gradient adjustments to bridge the gap between personalized and single-model federated learning techniques, advancing the efficiency and effectiveness of decentralized learning scenarios.

## 1 Introduction

Federated Learning (FL) is a compelling solution for constructing a shared (global) model from several local data sources that inherently cannot be exchanged or aggregated (Yang et al., 2019; Kairouz et al., 2021; Li et al., 2020b; Mahlool & Abed, 2022; Zhang et al., 2021). This is particularly essential in areas where data privacy or security is critical, with healthcare being a prime example (Li et al., 2020a; Rieke et al., 2020). FL operates through an iterative procedure involving rounds of model improvement until predefined criteria are met. These rounds include distributing the current global model to local entities (users) and selecting participants to contribute to the model update. The chosen users participate by training their local copies of the model using their respective data and communicating back the resulting model. The returned models are subsequently aggregated to create an updated global model. This process described above is the procedure for a typical single-model-based FL technique, where only one model is learned for all users. The most widely embraced single-model-based approach in FL is FedAvg (McMahan et al., 2017), which computes the new global model as the average of returned models.

When local data are not independent or identically distributed, ensuring good performance across different users with a single shared global model can be challenging (Qu et al., 2022; Caldarola et al., 2022). Recent works (Chen et al., 2022; Tan et al., 2022) argue that alternatively, the focus should be on personalized methods that output models adapted to the local data. However, learning personalized models requires additional memory and computation. When the data distributions of users are similar, a single model can perform well, and personalized techniques may waste resources to return a similar solution. A method that can perform well for heterogeneous user data but with a similar computational cost to traditional single-model-based federated learning techniques is desired.

Previous work by Nichol et al. (2018) finds that various meta-learning techniques achieve fixed balances between initial model success and rapid personalization due to different gradient strategies. Motivated by this finding, we propose a novel extension to FedAvg by scaling gradient updates

within local training. Our modification permits us to flexibly control this balance between initial model success and rapid personalization, which allows us to adapt to the data heterogeneity of user data in each application. Furthermore, despite being able to perform well on diverse local data sets, our training process requires a similar computational cost to FedAvg. The contribution of our work is summarized below:

- We proposed a novel algorithm (FedDecay) that improves the generalization performance of single-model-based federated learning techniques by incorporating gradient decay into local updates within each training round. Theoretical analysis demonstrates that FedDecay maintains the same order of convergence rate as FedAvg (Li et al., 2020c) under scenarios with non-independent and non-identically distributed local data sets.
- Empirical results spanning diverse data sets show that FedDecay enhances single-model-based techniques and bridges performance gaps with personalized federated learning methods without the additional computation and memory overhead required by such methods. Importantly, we consistently observe a 1 to 4 percentage point improvement in average test set accuracy for new and existing users over FedAvg.

The rest of the paper is organized as follows. First, Section 2 briefly summarizes related work from personalized federated learning and meta-learning. Second, Section 3 introduces our approach, mathematically justifies its improved flexibility, and establishes its convergence rate. Finally, Section 4 gives empirical evidence for performance improvements and computational cost compared with other benchmark methods.

## 2 RELATED WORK

**Personalized Federated Learning (pFL).** The simplest example of personalized federated learning involves fine-tuning the shared global model for each user after the federated learning process has terminated. More intricate solutions include learning a distinct model for each user (Li et al., 2021a), sharing a subset of the entire model globally (Li et al., 2021b), or treating user data as a mixture of distributions (Marfoq et al., 2021). Marfoq et al. (2021) can be considered an example of a strategy that clusters (or interpolates) users (Briggs et al., 2020; Ghosh et al., 2020; Mansour et al., 2020; Sattler et al., 2019) to reduce the additional models to learn. Regardless, learning other models throughout training requires substantial increases in memory and computation. Furthermore, several prevailing methods may prove restrictive or inequitable for applications acquiring new users. For instance, Ditto's approach (Li et al., 2021a) of training personalized models during federated learning is infeasible for non-participating users. Likewise, FedBN (Li et al., 2021b) and FedEM (Marfoq et al., 2021), relying on user data insights, pose challenges when accommodating new users lacking adequate data for fine-tuning.

**Meta-Learning.** A highly related perspective to personalized federated learning is meta-learning (Finn et al., 2017), focused on acquiring an effective model initialization to expedite convergence in downstream tasks. Importantly, sufficient data may need to be available as an initialization that converges quickly may not exhibit solid performance without fine-tuning. Although distinct in scope, personalized federated learning can be considered a subset of meta-learning, where an attempt is made to identify an initial model to perform well on diverse local data sets. Recent endeavors have applied meta-learning techniques to personalized federated learning (Jiang et al., 2019; Chen et al., 2018; Finn et al., 2017; Charles & Konečný, 2021). In addition to adapting conventional meta-learning algorithms to the federated learning context (Jiang et al., 2019), a variant named Reptile (Nichol et al., 2018) can be shown to be equivalent to FedAvg under the condition of equally sized local data sets. In this article, we only study aggregation with equally weighted users. Hence, we will use the term FedAvg to refer to both FedAvg and Reptile.

## 3 METHOD

We begin this section by introducing our notation for the federated learning process. Before proposing our method, we present the similarities and differences of the popular existing methods: FedSGD, FedAvg (McMahan et al., 2017), and FOMAML (Nichol et al., 2018). We believe this viewpoint of existing algorithms is essential for understanding the derivation and benefits of

our proposed method that follows. Finally, we mathematically justify that our proposed method can flexibly balance the goals of initial model success and rapid personalization and establish our convergence rate.

**Notation.** We consider federated learning conducted over $N$ communication rounds, each consisting of $K$ local update steps. The set $C = [1, \ldots, M]$ represents the users participating in federated learning, each with local objective function $F_i$, which federated learning aims to find the model (or models) that minimize the average local loss ($\frac{1}{M} \sum_{i=1}^{M} F_i$) over users. In each round, a subset of users, $S \subseteq C$, participates in the update of the global model, and $S = C$ is referred to as full user participation. The global model at communication round $n$ is denoted as $\theta_g^n$, and the $i$-th user's copy of the shared model after $k$ of $K$ planned local update steps with learning rate $\eta$ is represented as $\theta_i^{n,k}$. We omit $n$ for improved readability, using $\theta_g$ and $\theta_i^k$ when discussing only operations within a training round.

### 3.1 GLOBAL AND LOCAL UPDATES FOR FEDERATED LEARNING

We begin by defining $F_i^j$ as a sequence of loss functions of size $K$, denoted as $\{F_i^j\}_{j=0}^{K-1}$, where $K$ is the total number of planned local update steps. For instance, this sequence could correspond to evaluations of the local objective function on various mini-batches. We define the standard update equation for FedAvg or FedSGD procedure as follows:

$$\theta_i^k = \theta_i^{k-1} - \eta g_i^{k-1} \text{ for all } k = 1, \ldots, K \tag{1}$$

Here, $g_i^{k-1} = \nabla F_i^{k-1}(\theta_i^{k-1})$ represents the gradient of the local objective function during the $k$-th local update step. When $K > 1$, Equation 1 represents the update procedure of FedAvg. When $K = 1$, it reduces to FedSGD. For the case of full user participation ($S = C$), the locally updated models are aggregated as in Equation 2, which can also be written as the aggregation of changes made by local updates:

$$\theta_g \leftarrow \frac{1}{M} \sum_{i \in C} \theta_i^K = \theta_g + \frac{1}{M} \sum_{i \in C} \underbrace{(\theta_i^K - \theta_g)}_{\text{Method Gradient}} . \tag{2}$$

Here, $\theta_i^K$ represents the local model of the $i$-th user after the final local step ($K$). Using Equation 1 recursively, we can write $\theta_i^K = \theta_i^0 - \eta \times \sum_{j=0}^{K-1} g_i^j$. Noting $\theta_i^0 = \theta_g$, the change a user makes, $\theta_i^K - \theta_g$ is proportional to $\sum_{j=0}^{K-1} g_i^j$ when we ignore the learning rate $\eta$. We refer to the gradient component of a user's change as the gradient of the method, $g_{method}$. Different FL methods may use different gradients in the summation. Equation 3 presents the specific gradients used from Equation 1 for FedSGD, FedAvg (McMahan et al., 2017), and FOMAML (Nichol et al., 2018).

$$g_{FedSGD} = g_i^0, \; g_{FedAvg} = \sum_{j=0}^{K-1} g_i^j, \; g_{FOMAML} = g_i^{K-1} \tag{3}$$

### 3.2 FEDDECAY: GENERALIZING LOCAL UPDATES WITH GRADIENT DECAY

We propose FedDecay, which introduces scaled gradients within each local step to emphasize early gradients essential to initial model success. More rigorous justification for stressing early gradients is given in Section 3.3. However, recall that a single model should only be helpful when users have similar local data sets. In the case that users are alike, for performance, it would be reasonable to create a single data set from all local data sets for training. However, federated learning keeps training distributed to protect data privacy or to avoid some computational restrictions. Hence, the closest we could get to traditional machine learning is FedSGD, which approximates standard stochastic gradient descent while respecting the decentralized nature of the data. In summary, when users are similar and initial model success is desirable, the first gradient step used by FedSGD is crucial.

We propose using exponential decay to scale the gradients, which can be expressed as shown in Equation 4 and gives the gradients used by FedDecay in Equation 5. Observe that different than FedSGD, FedAvg, and FOMAML, which either fully include or remove some gradient terms from

the summation, our proposed method gives a more flexible weighting of each term. We recover FedSGD when $\beta = 0$ and FedAvg when $\beta = 1$. Hence, we can view our proposed method as a generalization. Our method is implemented by applying a locally decaying cyclic learning rate, with $\eta_t = \eta \times \beta^{(t \bmod K)}$, where iteration $t = nK + k$ (k local updates into round $n$).

$$\theta_i^k = \theta_i^{k-1} - \eta \beta^{(k-1)} g_i^{k-1} \text{ for all } k = 1, \ldots, K \tag{4}$$

$$g_{FedDecay} = \sum_{j=0}^{K-1} \beta^{(j)} g_i^j \tag{5}$$

### 3.3 BALANCING INITIAL MODEL SUCCESS AND RAPID PERSONALIZATION

To show the justification for FedDecay, we perform a Taylor analysis to demonstrate the trade-off between initial model success and personalization regarding the ratio between two quantities: Average Gradient (AvgGrad) and Average Gradient Inner product (AvgGradInner). Following the analysis in Nichol et al. (2018), AvgGrad and AvgGradInner are defined in the following part.

Delving deeper, let $\tilde{g}_i^{j-1} = \nabla F_i^{j-1}(\theta_g)$ and $\tilde{H}_i^{j-1} = \nabla^2 F_i^{j-1}(\theta_g)$ denote the gradient and Hessian, respectively, of the $j$-th loss function evaluated at the most recent global model. When we take expectation over the gradient of FedDecay, two quantities emerge: $\mathbb{E}_{i,k}[\tilde{g}_i^k]$ (AvgGrad) and $\mathbb{E}_{i,k,l}[\tilde{H}_i^k \tilde{g}_i^l]$ (AvgGradInner). Equation 6 shows the expectation of $g_{FedDecay}$ with exponential decay. See Section A for the full derivation, which includes more generalized $\beta$-sequences than just exponential decay.

$$\mathbb{E}\left[g_{FedDecay}\right] \approx \left(\frac{1 - \beta^K}{1 - \beta}\right) \mathbb{E}\left[\tilde{g}_i^k\right] - \left(\frac{(1 - \beta^{(K-1)})(1 - \beta^K)}{(1 + \beta)(1 - \beta)^2}\right) \mathbb{E}\left[\tilde{H}_i^k \tilde{g}_i^l\right] \tag{6}$$

The emphasis on AvgGrad encourages convergence toward joint training, minimizing the expected loss across users and achieving initial model success. Conversely, a negative focus on AvgGradInner maximizes the inner product between gradients (as illustrated in Equation 7 by the chain rule) from distinct mini-batches within the same user. Large inner products indicate that the gradient will recommend moving in a similar direction regardless of the input data. Hence, even a single update step can quickly improve the model performance on a large quantity of a user's data, thus facilitating rapid personalization.

$$\mathbb{E}_{i,k,l}\left[\tilde{H}_i^k \tilde{g}_i^l\right] = \frac{1}{2}\mathbb{E}\left[\tilde{H}_i^k \tilde{g}_i^l + \tilde{H}_i^l \tilde{g}_i^k\right] = \frac{1}{2}\mathbb{E}\left[\frac{\partial}{\partial \theta_g}\left(\tilde{g}_i^k \tilde{g}_i^l\right)\right] \tag{7}$$

When users possess similar data, a method that can be expressed as only AvgGrad terms can perform well across all users; however, in cases where users' data diverges significantly, an approach prioritizing AvgGradInner terms may be more suitable. Plausibly, most applications exist between the above two scenarios. In such cases, a method striking a flexible balance between AvgGrad and AvgGradInner terms could yield superior performance, including initial model success, rapid personalization, and generalization across a spectrum of statistical data heterogeneity. Please see Section D for experimental evidence for the above claims.

The exact quantities of interest (AvgGrad and AvgGradInner) also appear in the expected gradients of FedSGD, FedAvg, and FOMAML Nichol et al. (2018). By looking at the ratio of the coefficients of AvgGradInner to AvgGrad for various methods, we can assess each method's focus on initial model success vs. rapid personalization. See Table 1 to contrast the flexible ratio of FedDecay with those of FedSGD, FedAvg, and FOMAML. The ratios are fixed for FedSGD, FedAvg, and FOMAML, given $K$. On the other hand, the ratio of FedDecay depends on the choice of $\beta$.

Regardless of whether local data sets are similar, FedSGD, FedAvg, and FOMAML will place the same fixed emphasis on initial model success and rapid personalization. On the other hand, FedDecay with exponential decay emerges as the only method offering the flexibility to balance this intricate trade-off (with parameter $\beta$), particularly as the number of local steps increases (communication becomes less frequent). This additional flexibility allows FedDecay to adapt to the problem-specific data heterogeneity for better performance.

| Method | FedSGD | **FedDecay** | FedAvg | FOMAML |
|---|---|---|---|---|
| Ratio | $\dfrac{0}{1}$ | $\dfrac{\left(\beta - \beta^K\right)\eta}{1 - \beta^2}$ | $\dfrac{(K-1)\eta}{2}$ | $(K-1)\eta$ |
| Limit ($K \to \infty$) | $0$ | $\dfrac{\beta\eta}{1 - \beta^2}$ | $\infty$ | $\infty$ |

Table 1: Ratios of coefficients for AvgGradInner (rapid personalization) to AvgGrad (initial model success) terms for several methods. Choice of $\beta$ gives FedDecay much greater control over the ratio of emphasis on the ability to personalize vs. initial model success. Note that we are assuming that $\beta \in (0,1)$ for FedDecay since FedSGD ($\beta = 0$) and FedAvg ($\beta = 1$) are already present.

### 3.4 CONVERGENCE ANALYSIS

In this section, we establish that while we introduce greater flexibility by extending FedSGD and FedAvg to FedDecay, the core convergence properties of the algorithm remain strikingly similar, albeit with some constants involved.

Furthermore, we enhance FedDecay from its initial form in Equation 5 to a more versatile local update $\sum_{j=0}^{K-1} \beta_j g_i^j$, where $\beta_{j+1} \leq \beta_j$ to accommodate broader scenarios. While exponential decay (as discussed in Section 3.3) is a straightforward, intuitive, and mathematically sound choice, other types of decay can also emphasize early gradients and initial model success. We proceed to demonstrate that even with this more general decay approach, FedDecay converges with a complexity of $\mathcal{O}(N^{-1})$ for both full and partial user participation under the following assumptions.

In this section, $F_i$ is specifically the average local loss over the data of the $i$-th user. Please see Section B for additional details and proofs.

**Assumption 1** (Lipschitz Continuous Gradients). *For all users $i \in \{1, \ldots, M\}$, the local objective function $F_i$ is L-smooth. For all $v$ and $w$ in the domain of $F_i$,*

$$F_i(v) \leq F_i(w) + (v - w)^T \nabla F_i(w) + \frac{L}{2} \|v - w\|_2^2$$

**Assumption 2** (Strong Convexity). *For all users $i \in \{1, \ldots, M\}$, the local objective function $F_i$ is $\mu$-strongly convex. Similarly, $F_i(v) \geq F_i(w) + (v - w)^T \nabla F_i(w) + \frac{\mu}{2} \|v - w\|_2^2$*

**Assumption 3** (Bounded Variance). *Let $\xi^{(n,k)}$ be sampled uniformly from the $i$-th user's local data. For all $n$ and $k$, the variance of stochastic gradients in each user $i$ is bounded.*

$$\mathbb{E}\left\|\nabla \hat{F}_i\left(\theta_i^{(n,k)}, \xi_i^{(n,k)}\right) - \nabla F_i\left(\theta_i^{(n,k)}\right)\right\|^2 \leq \sigma_i^2$$

*where $\hat{F}_i$ is the average for $i$-th user's local loss when evaluated on data $\xi_i$.*

**Assumption 4** (Bounded Expected Squared Norm). *The expected squared norm of stochastic gradients is uniformly bounded. Similarly, $\mathbb{E}\left\|\nabla \hat{F}_i(\theta_i^{(n,k)}, \xi_i^{(n,k)})\right\|^2 \leq G^2$*

**Theorem 1** (Full User Participation). *Let Assumptions 1 to 4 hold and $L$, $\mu$, and $G$ be defined therein. Choose a locally decaying, cyclic learning rate for iteration $t = nK + k$ ($k$ local updates into round $n + 1$) of the form $\eta_t = \alpha_{\lfloor t/K \rfloor} \beta_{t \bmod K}$ for positive $\beta_j \geq \beta_{j+1}$ and some $\alpha_j = \frac{c}{j+d}$.*
*With $\kappa = \dfrac{L}{\mu}$, $d = \max\left\{\dfrac{12\kappa}{\beta_{K-1}}, 4 - \dfrac{2}{K}\right\}$ and $a_j = \dfrac{3}{\mu\beta_{K-1}(j+d)}$*

$$\mathbb{E}\left[F(\theta_g^N)\right] - F^* \leq \frac{\kappa}{N + d - 2}\left(\frac{9B}{2\mu\beta_{K-1}^2} + \frac{(1/K) + d - 2}{2} \times \mathbb{E}\left\|\theta_g^1 - \theta_g^*\right\|^2\right)$$

$$\text{where } B = \sum_{i=1}^{M} p_i^2 \sigma_i^2 + 6L\Gamma + 2\left(G(K-1)\beta_0\beta_{K-1}^{-1}\right)^2 \text{ and } \Gamma = F^* - \sum_{i=1}^{M} p_i F_i^*$$

**Theorem 2** (Partial User Participation). *Replacing the aggregation step with $\theta_g \leftarrow \frac{M}{|S|}\sum_{j \in S} p_j \theta_j^K$ where users $S \subset C$ of fixed size $|S|$ is sampled uniformly without replacement each round with*

*probabilities $p_i$ for user $i \in \{1, \ldots, M\}$. With $D = \frac{M-|S|}{|S|(M-1)}$, as in Theorem 1,*

$$\mathbb{E}\left[F(\theta_g^N)\right] - F^* \leq \frac{\kappa}{N+d-2}\left(\frac{9(B+D)}{2\mu\beta_{K-1}^2} + \frac{(1/K)+d-2}{2} \times \mathbb{E}\left\|\theta_g^1 - \theta_g^*\right\|^2\right)$$

The convergence analysis outlined in Theorem 1 and Theorem 2 emphasizes that the gap between the optimal solution and the global model produced by FedDecay converges to zero, akin to the outcomes seen with FedAvg, at rate $\mathcal{O}(N^{-1})$. However, the advantage of FedDecay lies in its ability to achieve this convergence while accommodating a spectrum of diverse updates. As illustrated in Section 3.3, incorporating local learning rate decay enables a versatile balance encompassing initial model success, rapid personalization, and generalization. Our research demonstrates that this enriched flexibility does not come at the expense of deteriorating algorithmic performance.

## 4 EXPERIMENTS

We emphasize transparency and reproducibility by providing open access to our experimental code and hyper-parameter search details[1]. In this section, we comprehensively evaluate our proposed method, FedDecay, alongside other prominent federated learning techniques, using the benchmark established by Chen et al. (2022). To ensure uniformity, we integrate our approach into their code base available at GitHub [2]. We leverage this established infrastructure to experiment with identical data sets and models to facilitate a rigorous and fair comparison.

**Benchmark Data Sets.** As discussed in Yuan et al. (2022), generalization in federated learning refers both to new users (participation gap) and new data for existing users (out-of-sample gap). To study both forms of generality, our experiments involve holding out both users and data in the following manner. For each data set, we maintain 20 percent of users as a holdout set, thereby simulating the challenge of generalization to new users. Also, each user's data is divided into distinct training, validation, and testing subsets. Our experiments encompass the various data sets FEMNIST (Caldas et al., 2018), SST2 (Wang et al., 2018; Socher et al., 2013), and PUBMED (Namata et al., 2012). This diversified selection of data sets empowers us to probe performance across varied domains, including vision, text, and graph data. We use the default data settings chosen by Chen et al. (2022).

- FEMNIST: A 62-way handwritten character classification problem with images of size $28 \times 28$. This sub-sampled version of FEMNIST in Chen et al. (2022) contains 43,400 images from 3,550 authors as clients.
- SST2: A sentiment classification data set containing 68,200 movie review sentences labeled with human sentiment. Partitioned into 50 clients using Dirichlet allocation with $\alpha = 0.4$, it enables the assessment of text-based applications.
- PUBMED: Comprising 19,717 nodes and 44,338 edges, this data set classifies scientific publications into three classes. It is partitioned into five clients using Louvain community partitioning.

**Model Architectures.** In our experiments, we adhere to the model architectures prescribed by Chen et al. (2022) for consistency and comparative purposes.

- FEMNIST: We employ a Convolutional Neural Network (CNN) with a hidden size of 2,048. The model incorporates two convolutional layers with $5 \times 5$ kernels, followed by max pooling, batch normalization, ReLU activation, and two dense layers.
- SST2: Our model leverages a pre-trained BERT-Tiny model (Turc et al., 2019) with 2-layer Transformer encoders and a hidden size of 128.
- PUBMED: We employ the Graph Isomorphism Neural Network (GIN) (Xu et al., 2019), featuring 2-layer convolutions with batch normalization, a hidden size of 64, and a dropout rate of 0.5.

---

[1]Code repository URL omitted for anonymous review

[2]Publicly available at `https://github.com/alibaba/FederatedScope/tree/Feature/pfl_bench`

**Baseline Methods.** To establish the prowess of FedDecay, we subject it to a rigorous comparison with a range of federated learning techniques.

- Personalized Methods: These encompass Ditto (Li et al., 2021a), FedBN (Li et al., 2021b), FedEM (Marfoq et al., 2021), and pFedMe (T Dinh et al., 2020).

- Single-Model-Based Methods: In this category, we include FedAvg/Reptile (McMahan et al., 2017; Nichol et al., 2018) and FOMAML (Nichol et al., 2018).

**Hyperparameter Configuration.** We extend the hyper-parameter optimization methodology outlined in the original benchmark paper. Leveraging a grid search technique, we conduct comprehensive explorations, leveraging early termination and hyperband stopping (Biewald, 2020). Our exhaustive search covers a spectrum of hyperparameters, as detailed in Table 2. The final hyperparameters for each run are selected based on the highest average validation accuracy across users. If runs are not terminated early, they will run for 1000, 500, and 500 epochs for FEMNIST, SST2, and PUBMED, respectively. Other fixed configurations worth noting are 20 percent partial user participation for FEMNIST and SST2, a batch size of 32 for FEMNIST, and full user participation with full-batch training for PUBMED. To ensure a fair comparison and alignment with other methods, we maintain a fixed learning rate ($\alpha$) for FedDecay, while $\beta$ is the key hyperparameter explored. Experiments are conducted on 4 NVIDIA GeForce RTX 3090 GPUs. See Section D to analyze FedDecay's performance under misspecified $\beta$.

| Hyper-parameter | Algorithm | Data Set | Tuning Grid |
|---|---|---|---|
| Local Epochs | - | - | $\{1, 3\}$ |
| Batch Size | - | SST2 | $\{16, 32, 64\}$ |
| Learning Rate | - | - | $\{0.005, 0.01, \ldots, 1.0\}$ |
| Regularization Rate | Ditto, pFedMe | - | $\{0.05, 0.1, 0.2, 0.5, 0.9\}$ |
| Meta-learning Step | pFedMe | - | $\{1, 3\}$ |
| Mixture Number | FedEM | - | $\{3\}$ |
| Local Decay | FedDecay | - | $\{0, 0.2, \ldots, 1.0\}$ |

Table 2: Hyper-parameter grid search details. If no algorithm or data set is specified, the given hyper-parameter search will be applied for all.

## 4.1 GENERALIZATION PERFORMANCE FOR NEW USERS

To assess the robustness of the methods in accommodating new users, we introduce the following performance metrics: $\bar{Acc}$ (average accuracy), $\breve{Acc}$ (bottom ten percentile accuracy), and $\sigma_{Acc}$ (standard deviation of accuracy). Average accuracy is the primary performance metric, and it is desirable to identify methods that often return large values for $\bar{Acc}$, indicating that the solution performs well for users, on average. In addition, we use the fairness metrics $\breve{Acc}$ and $\sigma_{Acc}$ to understand if solutions are performing well for many users. A fair method will consistently return a large bottom ten percentile accuracy and a low standard deviation, which indicate a lower bound on performance for the 90 percent majority of users and that the solution performs similarly across users, respectively. Please note that in the PUBMED data set comprising only five users, $\breve{Acc}$ and $\sigma_{Acc}$ are not applicable due to the limited number of held-out users. Additionally, we repeat the single-model-based results across multiple seeds in Section C to gain insights into result variability and partially assess the sensitivity of our findings to different initial conditions.

We initiate our analysis by investigating the proficiency of each method in catering to new users. The metrics are outlined in Table 3. Notably, our novel approach, FedDecay, outperforms all single-model-based methods regarding average test accuracy for all data sets. Furthermore, even in the FEMNIST data sets, it is worth highlighting that FedDecay significantly narrows the performance gap between single-model-based and personalized methods. While personalized methods such as FedEM and pFedMe exhibit competitive results on FEMNIST, the SST2 data set shows a noticeable drop in their performance. These findings underscore FedDecay's exceptional ability to generalize effectively to new users across diverse applications.

| Method | FEMNIST (image) | | | SST2 (text) | | | PUBMED (graph) | | |
|---|---|---|---|---|---|---|---|---|---|
| | $\bar{Acc}$ | $\breve{Acc}$ | $\sigma_{Acc}$ | $\bar{Acc}$ | $\breve{Acc}$ | $\sigma_{Acc}$ | $\bar{Acc}$ | $\breve{Acc}$ | $\sigma_{Acc}$ |
| Ditto | 0.5672 | 0.4444 | 0.0921 | 0.4746 | 0.0000 | 0.4010 | 0.2442 | - | - |
| FedBN | 0.9059 | 0.8302 | 0.0544 | **0.8030** | **0.6667** | 0.1275 | **0.8004** | - | - |
| FedEM | **0.9175** | 0.8333 | 0.0526 | 0.7404 | 0.6379 | 0.2141 | 0.7879 | - | - |
| pFedMe | 0.9036 | **0.8438** | 0.0751 | 0.7785 | 0.6406 | 0.1179 | 0.7932 | - | - |
| FOMAML | 0.8989 | 0.8113 | 0.0604 | 0.7680 | 0.6667 | 0.1057 | 0.7950 | - | - |
| FedAvg | 0.9055 | **0.8491** | 0.0530 | 0.7680 | 0.6667 | 0.1057 | 0.7914 | - | - |
| **FedDecay** | **0.9152** | 0.8421 | 0.0485 | **0.8101** | **0.6724** | 0.1087 | **0.8039** | - | - |

Table 3: Generalization to new users who did not participate in federated training after fine-tuning. FedDecay has the most considerable average test set accuracy ($\bar{Acc}$) of any single-model-based technique on all data sets, even outperforming personalized methods on SST2 and PUBMED. With five total users for PUBMED, only a single user is held out to evaluate generalization. Hence, there are no values for the bottom ten percentile ($\breve{Acc}$) or standard deviation ($\sigma_{Acc}$) for new users.

## 4.2 GENERALIZATION PERFORMANCE ON NEW DATA FOR EXISTING USERS

Turning our attention to the performance of users who participate in the federated learning process, we anticipate that personalized federated learning methods, characterized by additional memory and computation for learning personalized models, should outperform single-model techniques. The results, presented in Table 4, affirm this expectation. However, among the single-model-based methods, FedDecay emerges as the front-runner, achieving the highest average and bottom ten percentile test accuracy across all data sets. Furthermore, FedDecay even secures the highest average accuracy among all methods on the SST2 data set.

| Method | FEMNIST (image) | | | SST2 (text) | | | PUBMED (graph) | | |
|---|---|---|---|---|---|---|---|---|---|
| | $\bar{Acc}$ | $\breve{Acc}$ | $\sigma_{Acc}$ | $\bar{Acc}$ | $\breve{Acc}$ | $\sigma_{Acc}$ | $\bar{Acc}$ | $\breve{Acc}$ | $\sigma_{Acc}$ |
| Ditto | 0.9031 | 0.8333 | 0.0563 | 0.5949 | 0.0417 | 0.3449 | 0.8754 | 0.8465 | 0.0236 |
| FedBN | 0.9182 | 0.8571 | 0.0548 | 0.7360 | 0.4000 | 0.2292 | 0.8788 | **0.8540** | 0.0248 |
| FedEM | 0.8952 | 0.8200 | 0.0806 | **0.7808** | 0.5000 | 0.1845 | **0.8822** | 0.8478 | 0.0322 |
| pFedMe | **0.9280** | **0.8750** | 0.0694 | 0.7699 | **0.6087** | 0.1672 | 0.8666 | 0.8205 | 0.0334 |
| FOMAML | 0.8951 | 0.8140 | 0.0636 | 0.7654 | **0.5556** | 0.1634 | 0.8614 | 0.8292 | 0.0284 |
| FedAvg | 0.8851 | 0.8039 | 0.0750 | 0.7654 | **0.5556** | 0.1634 | 0.8671 | 0.8342 | 0.0267 |
| **FedDecay** | **0.8986** | **0.8214** | 0.0711 | **0.7815** | **0.5556** | 0.1727 | **0.8722** | **0.8490** | 0.0233 |

Table 4: Generalization to new data for existing users who participated in federated training after fine-tuning. FedDecay produces the best test set average and bottom ten percentile accuracy of any single-model-based method.

Delving into personalized federated learning techniques, we observe mixed results. Ditto and FedBN struggle on the SST2 data set, pFedMe faces challenges on PUBMED, and FedEM encounters difficulties on FEMNIST. Additionally, it is worth noting that FedDecay consistently outperforms other methods on at least one data set despite their additional computation and memory requirements. Please refer to Section 4.3 for a detailed exploration of the cost implications.

## 4.3 COMPUTATIONAL AND COMMUNICATION COSTS

The computational and communication costs incurred during training are integral to evaluating federated learning methods. Personalized federated learning methods, in particular, often demand increased computation and memory resources. In this context, we draw attention to the cost implications of various methods. As depicted in Figure 1, FedDecay exhibits comparable computational costs to other single-model-based techniques. The cost of learning rate scheduling is negligible per iteration, and the potential increase in communication rounds due to decaying updates doesn't meaningfully change computation. In our experiments, FedDecay returns an identical cost to FedAvg for FEMNIST and SST2, which terminate in the same number of epochs. FedDecay achieves similar or superior results without considerably higher communication and computation costs.

FedDecay requires substantially less memory and computation than FedEM and pFedMe, the two personalized methods that occasionally outperformed FedDecay in Section 4.1 and Section 4.2. Although FedDecay does not consistently outperform some personalized federated learning methods, it is much more computationally efficient, often dramatically closing the gap between other single-model-based methods and personalized federated learning solutions. This realization emphasizes the efficiency of FedDecay in striking a favorable balance between performance and resource utilization, making it an attractive solution for practical federated learning scenarios.

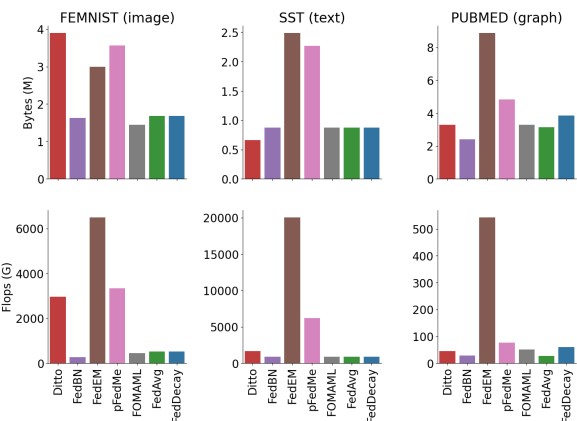

Figure 1: Total bytes communicated (*top*) and floating point operations (*bottom*) for each method's top-performing run. FedDecay produces similar costs to other single-model-based techniques. However, FedEM and pFedMe acquire much more significant costs.

## 5 LIMITATIONS

While our study offers valuable insights into the performance of our proposed method across various datasets, we acknowledge certain limitations that open avenues for future exploration and optimization. First, our analysis concentrated on a selection of data heterogeneity scenarios, encompassing natural and partitioned heterogeneity. We acknowledge that our findings may not fully encapsulate the dynamics present in more extreme non-iid scenarios.

Next, we focus mainly on exponential decay to show that scaling local gradient updates allows for balancing the objectives of initial model success and the ability to personalize rapidly. However, many alternative decay schemes for learning rates exist, but we have only briefly explored linear decay in Section B.3. Future research may explore these alternative decay schemes to enrich our understanding and potentially enhance our method's performance.

Finally, we evaluated our proposed method in isolation without extensively exploring its compatibility with other state-of-the-art techniques. While this approach enabled us to assess our method's intrinsic merits, practical deployment often requires combining multiple strategies for optimal results. Much of our theoretical work relies on stochastic gradient descent as the optimizer, and we intend to expand our theory to alternative optimizers in future work.

## 6 CONCLUSION

In summary, this study introduces an unexplored approach to federated learning by incorporating gradient decay into local updates within each round of training. FedDecay demonstrates remarkable performance improvements over other single-model-based techniques across vision, text, and graph data sets by dynamically adjusting the emphasis between initial model success and fine-tuning. Furthermore, FedDecay bridges the performance gap between single-model and personalized federated learning methods while avoiding the excessive computation and communication costs associated with the latter. Our work underscores the importance of tailored gradient adjustments in enhancing the personalization capabilities and efficiency of federated learning.

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

## A   BALANCING INITIAL MODEL SUCCESS AND RAPID PERSONALIZATION (CONTINUED)

We extend the work of Nichol et al. (2018) to understand how the updates of FedDecay impact initial model success, rapid personalization, and generalization. Consider the gradient from a sequence of loss functions $\{F_i^j\}_{j=0}^{K-1}$ where $K \in \{z \in \mathbb{Z} \mid z \geq 2\}$. For example, the above sequence could be a local objective function evaluated on different mini-batches. Consider the following update for any sequence of scaling coefficients $\{\beta_j \mid \beta_0 \neq 0\}_{j=0}^{K-1}$.

$$\theta_i^k = \theta_i^{k-1} - \eta \beta_{k-1} \times \nabla F_i^{k-1}(\theta_i^{k-1}) \text{ for } k = 1, \ldots, K$$

Let $g_i^j = \nabla F_i^j(\theta_i^j)$ and define $\tilde{g}_i^j = \nabla F_i^j(\theta_g)$ and $\tilde{H}_i^j = \nabla^2 F_i^j(\theta_g)$ as the gradient and Hessian of the $j$-th loss function evaluated at the initial point.

$$
\begin{aligned}
g_i^j = \nabla F_i^j(\theta_i^j) &\approx \nabla F_i^j(\theta_g) + \nabla^2 F_i^j(\theta_g)\left(\theta_i^j - \theta_g\right) \\
&= \tilde{g}_i^j + \tilde{H}_i^j\left(\theta_i^j - \theta_g\right) \\
&= \tilde{g}_i^j - \eta \tilde{H}_i^j \sum_{h=0}^{j-1} \beta_h \nabla F_i^h(\theta_i^h) \\
&\approx \tilde{g}_i^j - \eta \tilde{H}_i^j \sum_{h=0}^{j-1} \beta_h \tilde{g}_i^h
\end{aligned}
$$

Apply the previous expansion to the update of FedDecay.

$$
\begin{aligned}
g_{FedDecay} = \frac{\theta_i^K - \theta_g}{-\eta} &= \sum_{j=0}^{K-1}\left(\frac{\theta_i^{j+1} - \theta_i^j}{-\eta}\right) \\
&= \sum_{j=0}^{K-1} \beta_j \nabla F_i^j(\theta_i^j) \\
&\approx \sum_{j=0}^{K-1} \beta_j\left(\tilde{g}_i^j - \eta \tilde{H}_i^j \sum_{h=0}^{j-1} \beta_h \tilde{g}_i^h\right) \\
&= \sum_{j=0}^{K-1} \beta_j \tilde{g}_i^j - \eta \sum_{j=0}^{K-1}\left(\beta_j \tilde{H}_i^j \sum_{h=0}^{j-1} \beta_h \tilde{g}_i^h\right)
\end{aligned}
$$

$$
\begin{aligned}
\mathbb{E}\left[g_{FedDecay}\right] &\approx \sum_{j=0}^{K-1} \beta_j \tilde{g}_i^j - \eta \sum_{j=0}^{K-1}\left(\beta_j \tilde{H}_i^j \sum_{h=0}^{j-1} \beta_h \tilde{g}_i^h\right) \\
&= \mathbb{E}\left[\tilde{g}_i^j\right]\left(\sum_{j=0}^{K-1} \beta_j\right) - \mathbb{E}\left[\tilde{H}_i^j \tilde{g}_i^h\right]\left(\eta \sum_{j=0}^{K-1}\left(\beta_j \sum_{h=0}^{j-1} \beta_h\right)\right)
\end{aligned}
$$

Hence the following ratio of $\mathbb{E}\left[\tilde{H}_i^j \tilde{g}_i^h\right]$ to $\mathbb{E}\left[\tilde{g}_i^j\right]$ for FedDecay. Let $\mathcal{B}(k) = \sum_{j=0}^{k-1} \beta_{j-1}$.

$$R_{FedDecay} = \frac{\eta \sum_{j=0}^{K-1} \beta_j \mathcal{B}(j)}{\mathcal{B}(K)}$$

Exponential decay $\beta_j = \beta^j$ allows for the simplification of the above ratio after using the finite sum formula for the geometric series, $\mathcal{B}(k) = \frac{1-\beta^k}{1-\beta}$.

$$R_{FedDecay} = \eta \left( \sum_{j=0}^{K-1} \beta^j \left( \frac{1-\beta^j}{1-\beta} \right) \right) \times \left( \frac{1-\beta}{1-\beta^K} \right)$$

$$= \eta\beta \left( \frac{(1-\beta^K)(1-\beta^{(K-1)})}{(1-\beta)^2(1+\beta)} \right) \times \left( \frac{1-\beta}{1-\beta^K} \right)$$

$$= \eta\beta \left( \frac{1-\beta^{(K-1)}}{(1-\beta)(1+\beta)} \right)$$

$$= \eta\beta \left( \frac{1-\beta^{(K-1)}}{1-\beta^2} \right)$$

# B CONVERGENCE ANALYSIS (CONTINUED)

## B.1 FULL USER PARTICIPATION

We expand the theoretical work of Li et al. (2020c). Furthermore, we adopt the following additional notations and use their Lemma 1 and Lemma 2. Let $t$ denote the iteration of federated training. The $i$-th users model at iteration $t$ is provided by $w_t^i$. Note that with our notation $\lfloor t/K \rfloor = n$, the communication round, and $t \bmod K = k$ gives the local iteration step. Hence, all previous notations can be converted similarly to $w_t^i = \theta_i^{\lfloor t/K \rfloor, t \bmod K}$.

Recall that $S_t \subseteq C$ denotes the set of users participating in the update of the global model for a given iteration $t$ of federated training. In this section, $S. = C$ and let $p_i$ denote the aggregation weight of the $i$-th user. We are most interested in assigning equal probability or aggregation weight to all users since our objective is an improved initialization (for all users).

$$v_{t+1}^i = w_t^i - \eta_t \nabla \hat{F}_i(w_t^i, \xi_t^i)$$
$$w_{t+1}^i = \begin{cases} v_{t+1}^i & \text{if } (t+1) \bmod K \neq 0 \\ \sum_{i=1}^M p_i v_{t+1}^i & \text{else} \end{cases}$$

Also, we need to define the following aggregated sequences, which are always equivalent under full user participation. Furthermore, shorthand notation for various gradients is used for simplicity. Note $\bar{w}_t$ is only accessible when $t \bmod K = 0$, $\bar{g}_t = \mathbb{E}g_t$, and $\bar{v}_{t+1} = \bar{w}_t - \eta_t g_t$.

$$\bar{v}_t = \sum_{i=1}^M p_i v_t^i \qquad\qquad g_t = \sum_{i=1}^M p_i \nabla F_i(w_t^i, \xi_t^i)$$
$$\bar{w}_t = \sum_{i=1}^M p_i w_t^i \qquad\qquad \bar{g}_t = \sum_{i=1}^M p_i \nabla F_i(w_t^i)$$

We take the following two lemmas from Li et al. (2020c) and adopt the third to our learning rate scheduling:

**Lemma 1.** *Assume Assumption 1 and 2. If $\eta_t \leq \frac{1}{4L}$, we have*

$$\mathbb{E}\left\| \bar{v}_{(t+1)} - w^* \right\|^2 \leq (1 - \eta_t)\mathbb{E}\left\| \bar{w}_t - w^* \right\|^2 + \eta_t^2 \mathbb{E}\left\| g_t - \bar{g}_t \right\|^2$$
$$+ 2\mathbb{E}\left[ \sum_{i=1}^M p_i \left\| \bar{w}_t - w_i^t \right\|^2 \right] + 6L\eta_t^2 \Gamma$$

**Lemma 2.** *Under Assumption 3, it follows that*

$$\mathbb{E}\left\| g_t - \bar{g}_t \right\|^2 \leq \sum_{i=1}^M p_i^2 \sigma_i^2$$

**Lemma 3.** *Under Assumption 4, a locally decaying, cyclic learning rate of the form $\eta_t = \alpha_{\lfloor t/K \rfloor} \beta_{t \bmod K}$ such that $\beta_{t+1} \leq \beta_t$ for $t = 0, \ldots, K-1$ satisfies*

$$E\left[ \sum_{i=1}^M p_i \left\| \bar{w}_t - w_t^i \right\|^2 \right] \leq \eta_t^2 \left( G(K-1)\beta_{K-1}^{-1} \right)^2$$

*Proof.* Note that for all $t \geq 0$ there exists $t_0 \leq t$ such that $t - t_0 \leq K - 1$ and $w_{t_0}^i = \bar{w}_{t_0}$ for all $i \in [1, \ldots, M]$

$$\mathbb{E}\left[\sum_{i=1}^{M} p_i \left\|\bar{w}_t - w_t^i\right\|^2\right]$$

$$= \mathbb{E}\left[\sum_{i=1}^{M} p_i \left\|\left(w_t^i - \bar{w}_{t_0}\right) - \left(\bar{w}_t - \bar{w}_{t_0}\right)\right\|^2\right]$$

$$\leq \mathbb{E}\left[\sum_{i=1}^{M} p_i \left\|\left(w_t^i - \bar{w}_{t_0}\right)\right\|^2\right]$$

as $\mathbb{E}\|X - \mathbb{E}X\| \leq \mathbb{E}\|X\|^2$ where $X = w_t^i - \bar{w}_{t_0}$

$$\leq \sum_{i=1}^{M} p_i \mathbb{E}\left[(K-1)\sum_{i=t_0}^{t-1} \eta_t^2 \left\|\nabla \hat{F}_i\left(w_t^i, \xi_t^i\right)\right\|^2\right]$$

as $\left\|\sum_{i=t_0}^{t-1} \eta_t \nabla \hat{F}_i\left(w_t^i, \xi_t^i\right)\right\|^2 \leq (t - t_0)\sum_{i=t_0}^{t-1} \eta_t^2 \left\|\nabla \hat{F}_i\left(w_t^i, \xi_t^i\right)\right\|^2$

$$\leq (K-1)\sum_{i=1}^{M} p_i \sum_{i=t_0}^{t-1} \eta_{t_0}^2 G^2$$

as $\mathbb{E}\left\|\nabla \hat{F}_i\left(w_t^i, \xi_t^i\right)\right\|^2 \leq G^2$ and $\eta_t \leq \eta_{t_0}$ for $t_0 \leq t \leq t_0 + K$

$$\leq (K-1)\sum_{i=1}^{M} p_i \sum_{i=t_0}^{t-1} \left(\frac{G\beta_0}{\beta_{K-1}}\right)^2$$

as $\eta_{t_0} \leq \eta_t \left(\frac{\beta_0}{\beta_{K-1}}\right)$ for $t_0 \leq t \leq t_0 + K$

$$\leq \eta_t^2 \left(G(K-1)\beta_0 \beta_{K-1}^{-1}\right)^2 \text{ as } \sum_{i=1}^{M} p_i = 1$$

$\square$

### B.1.1 PROOF OF THEOREM 1

*Proof.* Recall that for all $t$, under full user participation $\bar{w}_t = \bar{v}_t$. Let $\Delta_t = \mathbb{E}\|\bar{w}_t - w^*\|^2$. From Lemma 1, 2, and 3, it follows that

$$\Delta_{t+1} \leq (1 - \eta_t \mu)\Delta_t + \eta_t^2 B$$

$$\text{where } B = \sum_{i=1}^{M} p_i^2 \sigma_i^2 + 6L\Gamma + 2\left(G(K-1)\beta_0 \beta_{K-1}^{-1}\right)^2$$

We can assume without loss of generality that $\beta_{t \bmod K} \leq 1$ and $\beta_0 = 1$. Otherwise

$$\eta_t = \alpha_{\lfloor t/K \rfloor}\beta_{t \bmod K} = \tilde{\alpha}_{\lfloor t/K \rfloor}\tilde{\beta}_{t \bmod K}$$

where $\tilde{\alpha}_t = \beta_0 \frac{c}{t+d}$ and $\tilde{\beta}_t = \frac{\beta_t}{\beta_0} \leq 1$. Additionally, we can assume that all $\beta$'s are positive-valued as we could reduce $K$ until this is satisfied.

For $\eta_t = \alpha_{\lfloor t/K \rfloor}\beta_{t \bmod K}$ where $\alpha_t = \frac{c}{t+d}$ for some $c > \frac{2}{\mu\beta_{K-1}}$ and $d > 2 - \frac{1}{K}$ such that $\eta_1 \leq \frac{1}{4L}$.

$$\frac{c}{(t/K) + d} \leq \alpha_{\lfloor t/K \rfloor} \leq \frac{c}{(t/K) + d - 1}$$

$$\Rightarrow \frac{c\beta_{K-1}}{(t/K) + d} \leq \eta_t \leq \frac{c}{(t/K) + d - 1}$$

$$\text{as } \beta_{K-1} \leq \beta_{t \bmod K} \leq 1$$

By induction, we prove $\Delta_t \leq \dfrac{v}{(t/K) + d - 2}$ where

$$v = \max \left\{ \frac{c^2 B}{\beta_{K-1} c \mu - 2}, \left[ (1/K) + d - 2 \right] \Delta_1 \right\}$$

Note that $t = 1$ holds trivially by the definition of $v$. Assuming the conclusion holds for some $t$, then

$$\Delta_{t+1} \leq (1 - \eta_t \mu) \Delta_t + \eta_t^2 B$$
$$\leq \left( 1 - \frac{\beta_{K-1} c \mu}{(t/K) + d} \right) \left( \frac{v}{(t/K) + d - 2} \right) + \left( \frac{c}{(t/K) + d - 1} \right)^2 B$$

by the induction hypothesis, $1 - \eta_t \mu \leq 1 - \dfrac{\beta_{K-1} c \mu}{(t/K) + d}$, and $\eta_t \leq \left( \dfrac{c}{(t/K) + d - 1} \right)$

$$\leq \left( 1 - \frac{\beta_{K-1} c \mu}{(t/K) + d} \right) \left( \frac{v}{(t/K) + d - 2} \right) + \frac{c^2 B}{[(t/K) + d] \, [(t/K) + d - 2]}$$
$$\text{as } (t/K) + d - 2 > (1/K) + 2 - (1/K) - 2 = 0$$
$$\Rightarrow [(t/K) + d] \, [(t/K) + d - 2] \leq [(t/K) + d - 1]$$

Letting $a = \dfrac{1}{[(t/K) + d] \, [(t/K) + d - 2]}$, we continue with the previous quantity.

$$= a \left( [(t/K) + d - 2] \, v + c^2 B - [c \mu \beta_{K-1} - 2] \, v \right)$$
$$\leq a \left( [(t/K) + d - 2] \, v + c^2 B - [c \mu \beta_{K-1} - 2] \left[ \frac{c^2 B}{\beta_{K-1} c \mu} \right] \right)$$

$$\text{as } \beta_{K-1} c \mu - 2 > 0 \text{ and } v \leq \frac{c^2 B}{\beta_{K-1} c \mu}$$

$$= \frac{[(t/K) + d - 2] \, v}{[(t/K) + d] \, [(t/K) + d - 2]}$$
$$= \frac{v}{(t/K) + d}$$
$$\leq \frac{v}{(t/K) + d} \left( \frac{(t/K) + d}{(t/K) + d - \left( 2 + \frac{1}{K} \right)} \right)$$
$$= \frac{v}{(t/K) + d - \left( 2 + \frac{1}{K} \right)}$$
$$= \frac{v}{\left( \frac{t+1}{K} \right) + d - 2}$$

Then by $L$-smoothness of $F(\cdot)$

$$\mathbb{E} \left[ F(\bar{w}_t) - F^* \right] \leq \frac{L}{2} \Delta_t \leq \frac{L}{2} \left( \frac{v}{(t/K) + d - 2} \right)$$

Specifically, if we choose the constants to be set as:

$c = \dfrac{3}{\mu\beta_{K-1} - 2}$ and $d = \max\left\{\dfrac{12L}{\mu\beta_{K-1}}, 4 - \dfrac{2}{K}\right\}$ then,

$$\alpha_1 = \frac{c}{d+1} < \frac{c}{d} = \frac{3}{\mu\beta_{K-1}d} \leq \frac{3}{12L + \mu\beta_{K-1}} \leq \frac{1}{4L}$$

$$\text{as } d \geq \frac{12L}{\mu\beta_{K-1}}$$

$$v = \max\left\{\frac{c^2 B}{\beta_{K-1}c\mu - 2}, [(1/K) + d - 2]\,\Delta_1\right\}$$

$$\leq \frac{c^2 B}{\beta_{K-1}c\mu - 2} + [(1/K) + d - 2]\,\Delta_1$$

$$= c^2 B + [(1/K) + d - 2]\,\Delta_1$$

$$\text{as } c = \frac{3}{\mu\beta_{K-1}}$$

$$= \left(\frac{3}{\mu\beta_{K-1}}\right)^2 B + [(1/K) + d - 2]\,\Delta_1$$

$$= \frac{2}{\mu}\left(\frac{9B}{2\mu\beta_{K-1}^2} + \frac{(1/K) + d - 2}{2}\Delta_1\right)$$

$$\mathbb{E}\left[F(\bar{w}_t) - F^*\right] \leq \frac{L}{2}\left(\frac{v}{(t/K) + d - 2}\right)$$

$$\leq \frac{\kappa}{(t/K) + d - 2}\left(\frac{9B}{2\mu\beta_{K-1}^2} + \frac{(1/K) + d - 2}{2}\Delta_1\right)$$

$\square$

## B.2 PARTIAL USER PARTICIPATION

Here, we focus on the case where the random set of users at iteration $t$ ($S_t$) of size $|S|$ is selected to update the global model. Consider when the central server forms $S.$ through sampling uniformly without replacement. We modify the definition of $w_t$ to incorporate the new averaging scheme.

$$w_{t+1}^i = \begin{cases} v_{t+1}^i & \text{if } (t+1) \bmod K \neq 0 \\ \sum_{i \in S_{t+1}} p_i \frac{M}{|S|} v_{t+1}^i & \text{else} \end{cases}$$

We rely on the following previous work by Li et al. (2020c).

**Lemma 4.** *If $t+1 \bmod K = 0$ and $S.$ is sampled uniformly without replacement, then*
$$\mathbb{E}_{S_t}(\bar{v}_{t+1}) = \bar{v}_{t+1}$$

**Lemma 5.** *If $t+1 \bmod K = 0$, $S.$ is sampled uniformly without replacement, $p_i = \frac{1}{n}$ for all $i \in [1, \ldots, M]$, and $\eta_t = \alpha_{\lfloor t/K \rfloor} \beta_{t \bmod K}$ where both $\alpha_j$ and $\beta_j$ are non-increasing, then*

$$\mathbb{E}_{S_t} \|\bar{v}_{t+1} - \bar{w}_{t+1}\|^2 \leq \eta_t^2 \left( \frac{M - |S_t|}{K(M-1)} \right) (GK\beta_0\beta_{K-1})^2$$

*Proof.* Replace $\eta_{t_0} \leq 2\eta_t$ with $\eta_{t_0} \leq \eta_t \left( \frac{\beta_0}{\beta_{K-1}} \right)$ in the original proof. $\qquad \square$

### B.2.1 PROOF OF THEOREM 2

*Proof.*

$$\begin{aligned}
&\mathbb{E}_{S_{t+1}} \|\bar{w}_{t+1} - w^*\|^2 \\
&= \mathbb{E}_{S_{t+1}} \|\bar{w}_{t+1} - \bar{v}_{t+1} + \bar{v}_{t+1} - w^*\|^2 \\
&= \mathbb{E}_{S_{t+1}} \|\bar{w}_{t+1} - \bar{v}_{t+1}\|^2 + \mathbb{E}_{S_{t+1}} \|\bar{v}_{t+1} - w^*\|^2 + 2\mathbb{E}_{S_{t+1}} \langle \bar{w}_{t+1} - \bar{v}_{t+1}, \bar{v}_{t+1} - w^* \rangle \\
&= \mathbb{E}_{S_{t+1}} \|\bar{w}_{t+1} - \bar{v}_{t+1}\|^2 + \mathbb{E}_{S_{t+1}} \|\bar{v}_{t+1} - w^*\|^2
\end{aligned}$$

The last equality follows from Lemma 4. Next, by Lemma 5, we have

$$\mathbb{E} \|\bar{w}_{t+1} - w^*\|^2 \leq \begin{cases} (1 - \eta_t\mu)\mathbb{E} \|\bar{w}_t - w^*\|^2 + \eta_t B \\ \qquad \text{if } t+1 \bmod K \neq 0 \\ (1 - \eta_t\mu)\mathbb{E} \|\bar{w}_t - w^*\|^2 + \eta_t(B + D) \\ \qquad \text{if } t+1 \bmod K = 0 \end{cases}$$

Furthermore, the second bound holds for all $t$ since $D > 0$. Next, similar to our proof under full user participation:

- Assume without loss of generality that $\beta_{t \bmod K} \leq 1$ and $\beta_0 = 1$.

- $\eta_t = \alpha_{\lfloor t/K \rfloor} \beta_{t \bmod K}$ where $\alpha_j = \frac{c}{t+d}$ for some $c > \frac{2}{\mu\beta_{K-1}}$ and $d > 2 - \frac{1}{K}$.

Then $\Delta_t \leq \dfrac{v}{(t/K) + d - 2}$ where $v = \max \left\{ \dfrac{c^2(B+D)}{\beta_{K-1}c\mu - 2}, [(1/K) + d - 2]\Delta_1 \right\}$.

Hence $\mathbb{E}[F(\bar{w}_t)] - F^* \leq \dfrac{L}{2}\Delta_t \leq \dfrac{L}{2} \left( \dfrac{v}{(t/K) + d - 2} \right)$.

With $c = \dfrac{3}{\mu\beta_{K-1}}$ and $d = \max \left\{ \dfrac{12L}{\mu\beta_{K-1}}, 4 - \dfrac{2}{K} \right\}$,

$$\mathbb{E}[F(\bar{w}_t) - F^*] \leq \frac{\kappa}{(t/K) + d - 2} \left( \frac{9(B+D)}{2\mu\beta_{K-1}^2} + \left( \frac{(1/K) + d - 2}{2} \right) \Delta_1 \right)$$

$\qquad \square$

## B.3 EXPLORING DIFFERENT DECAY SCHEMES

In this section, we explore the potential of alternative decay schemes within the FedDecay framework. While exponential decay has been a focus, we believe other strategies for scaling gradient emphasis during federated training could enhance performance by optimizing the balance between initial model success, generalization, and rapid personalization.

We expand the analysis to compare the experimental results of FedDecay using exponential and linear decay schemes. Linear decay is defined as $\beta_j = \max\{1 - j(1 - \beta), 0\}$, where $1 - \beta$ is employed for consistency aligning with FedSGD ($\beta = 0$) and FedAvg ($\beta = 1$). Figure 2 illustrates the average test set accuracy for new and existing users across different decay schemes.

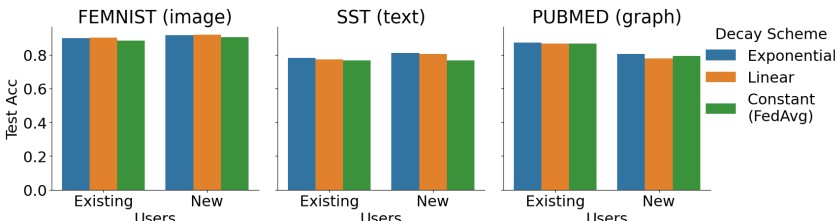

Figure 2: Average test set accuracy for new and existing users produced by FedDecay with different decay schemes compared with FedAvg. Either decay scheme with a simple grid search leads to performance improvements for all users on FEMNIST and SST2. Only linear decay on the new PUBMED user performs worse than FedAvg, but recall there is only one held-out user.

It is evident from the results that both exponential and linear decay schemes yield performance improvements over FedAvg. While linear decay performs slightly worse than FedAvg on the new PUBMED user, it demonstrates robust performance on FEMNIST and SST2. These findings underscore the potential for alternative decay schemes to enhance the overall model performance within the federated learning paradigm.

## C  ROBUSTNESS TO SEED CHOICE

To ensure the robustness of our findings against the selection of random seeds, we address this concern by conducting a set of experiments. Given the constraints of our computational resources, we focus on replicating the less computational single-model-based methods using three distinct fixed seeds. We chose FEMNIST and PUBMED to have one example of full and partial user participation. We present the averaged top-run metrics for new and existing users across the FEMNIST and PUBMED data sets in Table 5 and Table 6. Notably, the FedDecay approach consistently achieves the highest average accuracy among the single-model-based algorithms, catering to new and existing users. Moreover, it demonstrates exceptional performance regarding the best ten-percentile accuracy for existing users. This reinforces the reliability and generalizability of our proposed FedDecay method.

| Method | Existing Users | | | New Users | | |
|---|---|---|---|---|---|---|
| | $\bar{Acc}$ | $\breve{Acc}$ | $\sigma_{Acc}$ | $\bar{Acc}$ | $\breve{Acc}$ | $\sigma_{Acc}$ |
| FOMAML | 0.8971 | 0.8223 | 0.0687 | 0.9044 | **0.8371** | 0.0578 |
| FedAvg | 0.8948 | 0.8227 | 0.0701 | 0.9095 | 0.8318 | 0.0563 |
| **FedDecay** | **0.8993** | **0.8285** | 0.0688 | **0.9127** | 0.8294 | 0.0545 |

Table 5: Generalization metrics for new and existing users on the FEMNIST (image) data set. Values reported are the average metric for each single-model-based method's top run across several seeds. FedDecay produces the maximum average test set accuracy on new and existing users.

| Method | Existing Users | | | New Users | | |
|---|---|---|---|---|---|---|
| | $\bar{Acc}$ | $\breve{Acc}$ | $\sigma_{Acc}$ | $\bar{Acc}$ | $\breve{Acc}$ | $\sigma_{Acc}$ |
| FOMAML | 0.8597 | 0.8234 | 0.0293 | 0.7855 | - | - |
| FedAvg | 0.8666 | 0.8313 | 0.0328 | 0.7914 | - | - |
| **FedDecay** | **0.8704** | **0.8420** | 0.0244 | **0.7950** | - | - |

Table 6: Generalization metrics for new and existing users on the PUBMED (graph) data set. Values reported are the average metric for each single-model-based method's top run across several seeds. FedDecay produces the maximum average test set accuracy on new and existing users. With five total users, only a single user is held out to evaluate generalization. Hence, there are no values for the bottom ten percentile ($\breve{Acc}$) or standard deviation ($\sigma_{Acc}$) for new users.

# D  SENSITIVITY ANALYSIS

Our method demonstrates sensitivity to the hyper-parameter $\beta$ choice; see Table 7. Specifically, in the case of the FEMNIST data set, runs with $\beta = 0.8$ were prematurely halted due to hyper-band stopping, indicating inferior performance compared to smaller $\beta$ choices. Conversely, the remaining $\beta$ values improved accuracy for new and existing users. Notably, the FEMNIST data set, comprised of handwritten characters contributed by different authors, showcases inherent data heterogeneity. This real-world example underscores the presence of substantial user similarities, owing to the shared language in which all characters are written.

| Method | $\beta$ | FEMNIST (image) | | SST2 (text) | | PUBMED (graph) | |
|---|---|---|---|---|---|---|---|
| | | Existing $\bar{Acc}$ | New $\bar{Acc}$ | Existing $\bar{Acc}$ | New $\bar{Acc}$ | Existing $\bar{Acc}$ | New $\bar{Acc}$ |
| FedSGD | 0.0 | 88.51 | 90.55 | 73.60 | 80.30 | 86.70 | 79.14 |
| FedDecay | 0.2 | **89.86** | **91.52** | 67.01 | 59.02 | 85.67 | 79.68 |
| FedDecay | 0.4 | 89.74 | 90.77 | 69.42 | 67.26 | **87.22** | **80.39** |
| FedDecay | 0.6 | 89.76 | 90.93 | **78.15** | **81.01** | 86.65 | 79.86 |
| FedDecay | 0.8 | Hyperband Stopped | | 74.64 | 76.14 | 86.59 | 79.32 |
| FedAvg | 1.0 | 87.91 | 89.78 | 76.54 | 76.80 | 86.29 | 79.50 |

Table 7: Performance of FedDecay under misspecified values for decay coefficient, $\beta$.

This aligns with our assertion in Section 3.3, where we indicated that prioritizing AvgGrad terms (via smaller $\beta$ values) would be beneficial in cases where users share similarities. Consequently, we observe FedSGD outperforming FedAvg in this context. However, FedSGD, which does not emphasize AvgGradInner (generalization, rapid personalization), is beaten by FedDecay. Here, we observe that having a more flexible balance on initial model success, generalization, and fast personalization results in better performance for various choices of $\beta$.

Unlike FEMNIST, heterogeneous users are created from SST2 by partitioning data into 50 clients using Dirichlet allocation with $\alpha = 0.4$. Many extreme non-iid data sets exist, but $\alpha = 0.4$ results in reasonably non-iid users. Revisiting the claim discussed in our Section 3.3, we anticipate that an increased emphasis (achieved via larger $\beta$ values) on AvgGradInner is required as non-iid user diversity grows. In general, larger values of $\beta$ perform better on SST2. While FedSGD demonstrates superior test-set performance compared to FedAvg, its validation-set accuracy lags behind FedAvg's, preventing it from securing the best run on SST2 for FedAvg in Section 4.1. More importantly, FedDecay balances AvgGrad and AvgGradInner terms, yielding significantly improved metrics over FedAvg. Notably, FedDecay still has the best overall performance among the three methods.

Lastly, in the case of the PUBMED data set, our method's performance seems resilient to the choice of $\beta$. However, a straightforward grid search enhances performance for new and existing users over FedAvg and FedSGD.

# E  HYPERPARAMETER DETAILS

We provide the hyperparameter configurations for all methods that produce the top average validation set accuracy. These hyperparameter configurations are then used to compute test set metrics for our main experiments.

| Method | Local Epochs ($K$) | Batch Size | Learning Rate ($\alpha$) | Regularization Rate | Meta-learning Steps | Local Decay ($\beta$) |
|---|---|---|---|---|---|---|
| Ditto | 3 | - | 0.10 | 0.50 | - | - |
| FedBN | 3 | - | 0.01 | - | - | - |
| FedEM | 3 | - | 0.10 | - | - | - |
| pFedMe | 3 | - | 0.50 | 0.05 | 3.0 | - |
| FOMAML | 3 | - | 0.50 | - | - | - |
| FedAvg | 1 | - | 0.10 | - | - | - |
| FedDecay - Exponential | 3 | - | 0.05 | - | - | 0.2 |
| FedDecay - Linear | 3 | - | 0.10 | - | - | 0.6 |

Table 8: Main experiment hyper-parameters for all methods after tuning on FEMNIST

| Method | Local Epochs ($K$) | Batch Size | Learning Rate ($\alpha$) | Regularization Rate | Meta-learning Steps | Local Decay ($\beta$) |
|---|---|---|---|---|---|---|
| Ditto | 3 | 16.0 | 0.05 | 0.8 | - | - |
| FedBN | 3 | 64.0 | 0.10 | - | - | - |
| FedEM | 3 | 16.0 | 0.05 | - | - | - |
| pFedMe | 3 | 64.0 | 0.05 | 0.8 | 3.0 | - |
| FOMAML | 3 | 16.0 | 0.05 | - | - | - |
| FedAvg | 3 | 16.0 | 0.05 | - | - | - |
| FedDecay - Exponential | 3 | 16.0 | 0.05 | - | - | 0.6 |
| FedDecay - Linear | 3 | 16.0 | 0.05 | - | - | 0.4 |

Table 9: Main experiment hyper-parameters for all methods after tuning on SST

| Method | Local Epochs ($K$) | Batch Size | Learning Rate ($\alpha$) | Regularization Rate | Meta-learning Steps | Local Decay ($\beta$) |
|---|---|---|---|---|---|---|
| Ditto | 1 | - | 0.005 | 0.1 | - | - |
| FedBN | 3 | - | 0.005 | - | - | - |
| FedEM | 3 | - | 0.500 | - | - | - |
| pFedMe | 3 | - | 0.500 | 0.5 | 1.0 | - |
| FOMAML | 3 | - | 0.500 | - | - | - |
| FedAvg | 1 | - | 0.500 | - | - | - |
| FedDecay - Exponential | 3 | - | 0.500 | - | - | 0.4 |
| FedDecay - Linear | 3 | - | 0.500 | - | - | 0.4 |

Table 10: Main experiment hyper-parameters for all methods after tuning on PUBMED

