# OpenReview forum: "FedDecay: Adapting to Data Heterogeneity in Federated Learning With Gradient Decay"
_ICLR.cc/2024/Conference — ICLR 2024 Conference Withdrawn Submission_

### Official Review · Reviewer_wCRQ · 2023-10-18

**Soundness:** 2 fair
**Presentation:** 2 fair
**Contribution:** 1 poor
**Rating:** 3
**Confidence:** 4

**Summary:**

The paper proposes a novel cyclic decaying learning rate scheduling for federated learning that puts more emphasise on updates at the beginning of each communication round. That is, the learning rate for each communication round decays up until the next communication. The paper shows convergence for strongly convex, Lipschitz local objectives with bounded gradients and bounded gradient variance. The empirical evaluation shows that this learning rate scheduling achieves results en-par with other state-of-the-art federated learning approaches (FedBN, FedEM, and pFedME) both for standard FL settings and personalized FL settings.

**Strengths:**

- Sound motivation for the learning rate scheduling.
- Generalization in heterogeneous federated scenarios is a particularly challenging and relevant topic.
- Sound convergence proof of FedDecay.
- Details for reproducibility are provided.
- FedDecay is compared with some state-of-the-art algorithms and shows similar performances on the chosen benchmarks in terms of accuracy, computational and communication costs.
- Clearly stated motivation and discussion of some limitations.

**Weaknesses:**

- Very limited contribution.
- Limited advantages over existing methods.
- The problems of federated learning and personalized federated learning are not rigorously defined. The connection between heterogeneous data (often interpreted as data being drawn non-iid from a global distribution) and personalized federated learning (often defined as collaborative learning of local models with local data drawn from different, but related distributions) is not made clear.
- The paper lacks discussion of many relevant papers on heterogeneous federated learning [1,2,3,4,5,6,7,8].
- The theoretical analysis is very close to [9], which already discusses the necessity of decaying learning rates. In particular, this paper does not explicitly state that a globally decaying learning rate is required, but smuggles this assumption in through the definition of $\alpha_j$. So ultimately, the learning rate is not cyclic decaying, but the product of a cyclic decaying learning rate and a linearly decaying learning rate. It is not surprising that the multiplication with some cyclic decaying learning rate does not change the order of convergence. Thus, I do not see the contribution, here.
- The empirical evaluation does not convincingly show an advantage of the proposed method. The method performs en par with other methods in terms of accuracy, communication and computation.
- The paper does not evaluate different degrees of heterogeneity of local datasets, or using different local data distributions (as common in PFL).



[1] Li, Tian, et al. "Federated optimization in heterogeneous networks." Proceedings of Machine learning and systems 2 (2020): 429-450.
[2] Karimireddy, Sai Praneeth, et al. "Scaffold: Stochastic controlled averaging for federated learning." International conference on machine learning. PMLR, 2020.
[3] Adilova, Linara, et al. "FAM: Relative Flatness Aware Minimization." Topological, Algebraic and Geometric Learning Workshops 2023. PMLR, 2023.
[4] Durmus Alp Emre Acar, Yue Zhao, Ramon Matas Navarro, Matthew Mattina, Paul N Whatmough, and Venkatesh Saligrama. Federated learning based on dynamic regularization. International Conference on Learning Representations, 2021.
[5] Reddi, Sashank, et al. "Adaptive federated optimization." ICLR (2021).
[6] Li, Qinbin, Bingsheng He, and Dawn Song. "Model-contrastive federated learning." Proceedings of the IEEE/CVF conference on computer vision and pattern recognition. 2021.
[7] Karimireddy, Sai Praneeth, et al. "Mime: Mimicking centralized stochastic algorithms in federated learning." Advances in Neural Information Processing Systems, 2021.
[8] Wang, Jianyu, et al. "Slowmo: Improving communication-efficient distributed sgd with slow momentum." ICLR (2020).
[9] Li, Xiang, et al. "On the convergence of fedavg on non-iid data." ICLR (2020).

**Questions:**

- Could the cyclic learning rate be used to promote personalization, e.g., by putting more emphasize on later updates in each cycle?
- From Appendix B.3 it seems that different decay schemes have no substantial impact on the results. Could you get more substantial differences on specifically designed synthetic data? If so, could you from this argue for which type of data this method is applicable?

---

### Official Review · Reviewer_THnL · 2023-11-01

**Soundness:** 2 fair
**Presentation:** 2 fair
**Contribution:** 1 poor
**Rating:** 3
**Confidence:** 3

**Summary:**

The paper introduces FedDecay to handle heterogeneous federated scenarios by learning a global model more capable at generalizing.  FedDecay decays the magnitude of gradients during local training to balance personalization and generalization to the overall data distribution. FedDecay is shown to converge under both full and partial client participation.
Experimental evaluation is performed on vision, text and graph datasets, comparing generalization performance on both new users and unseen data of training clients.

**Strengths:**

- The paper addresses a relevant issue for the FL community, i.e. generalization under statistical heterogeneity
- Most of the relevant related works are discussed
- FedDecay is shown to converge under full and partial participation in convex settings.
- Experiments are run on different tasks (image classification, sentiment classification, graph classification), showing promising results against state-of-the-art baselines. Details for reproducing the experiments are provided, together with necessary computational resources.
- Limitations are explicitly addressed.

**Weaknesses:**

- The paper does not discuss previous studies addressing learning rate decay in FL. Examples are [1][2][3][4]. How is this work placed w.r.t. such studies?
- My main concern regards the novelty introduced by this work. As mentioned above, previous works discuss learning rate decay in local federated training. Specifically, [4] shows the necessity to leverage learning rate decay to reach fast convergence in $\mathcal{O}(\frac{1}{N})$ and consequent better generalization. How does FedDecay differ from these findings?
- Missing convergence proof under non-convex settings.
- In my opinion, the experimental settings are not complete. For instance, different client participation (partial vs full participation) is not analyzed, as well as the impact of different degrees of data heterogeneity. In addition, $\alpha=0.4$ for SST2 does not create a strongly heterogeneous setting and this choice is not discussed.
- The performance gains introduced in Table 1 and 2 are minimal.
- The parameter $\alpha$ of FedDecay is kept constant and not further analyzed. While it is reasonable to keep it fixed when comparing with other algorithms, it would be interesting to understand its impact and eventual connection with $\beta$.


**References**
[1] Cho, Yae Jee, Jianyu Wang, and Gauri Joshi. "Client selection in federated learning: Convergence analysis and power-of-choice selection strategies." arXiv preprint arXiv:2010.01243 (2020).
[2] Yu, Hao, Sen Yang, and Shenghuo Zhu. "Parallel restarted SGD with faster convergence and less communication: Demystifying why model averaging works for deep learning." AAAI. (2019)
[3] Li, Zexi, et al. "Revisiting weighted aggregation in federated learning with neural networks." ICML (2023).
[4] Li, Xiang, et al. "On the convergence of fedavg on non-iid data." ICLR (2020).

**Questions:**

1. How does FedDecay behave when faced with more extreme data distributions? And is FedDecay effective when faced with small client participation rates, e.g. 5%?
2. Since the paper addresses the issue of generalization, I believe it would be interesting to look at the results not only from the personalization, but also from the global one. How does FedDecay compare with methods proper of the literature of heterogeneous FL, e.g. SCAFFOLD, FedDyn, FedSAM?
3. Communication is the main bottleneck in FL. How does FedDecay compare with the other methods in terms of convergence speed?
4. Does the gradient decay introduced by FedDecay help reduce the client drift? Is the local overfitting reduced?

---

### Official Review · Reviewer_kknZ · 2023-11-04

**Soundness:** 4 excellent
**Presentation:** 3 good
**Contribution:** 3 good
**Rating:** 6
**Confidence:** 4

**Summary:**

In this paper, the researchers propose a novel algorithm (FedDecay) to improve the generalization performance of single-model-based federated learning techniques by incorporating gradient decay into local updates within each training round. The proposed method is shown effective for improving performance of single-model-based federated learning algorithms and is more efficient than the personalized federated learning methods.

**Strengths:**

1. The proposed FedDecay algorithm is very simple yet is theoretically a generalization of existing methods like FedSGD and FedAvg which can achieve more flexible weighting of the gradient terms.
2. This paper gives a deep insight into the single-model-based federated learning algorithm from the perspective of balancing initial model success and rapid personalization, which explains why FedDecay can adapt to the data heterogeneity for improving the performance.
3. Experimental results show that FedDecay outperforms existing single-model-based federated learning algorithms on three datasets of different domains and is more efficient than the personalized federated learning methods.

**Weaknesses:**

1. The memory overhead is mentioned in Section 1, but I cannot see experimental results about it.
2. The theoretical analysis of the efficiency (e.g., time, space and communication complexity) is generally lacked.
3. FedDecay is sensitive to the introduced hyper-parameter \beta and thus hyper-parameter tuning is needed to achieve considerable performance, which will cause additional computation and communication cost.
4. Some issues need further clarification, see questions.

**Questions:**

1. Provide theoretical and experimental analysis of the efficiency of the proposed method, including computation, communication and memory overhead.
2. While Figure 1 shows the total communicated and floating point operations for the top-performing run, I would like to see how the model performs with respect to each round of training because the number of rounds is usually a bottleneck of the communication efficiency for federated learning.
3. More clarification and justification are needed to show whether the proposed FedDecay is superior to the personalized federated learning method in consideration of the effort for tuning the hyper-parameter \beta.
4. What the term “initial model success” means in Section 3.3? Please clarify. And the mechanism that “Large inner products indicate that the gradient will recommend moving in a similar direction regardless of the input data. Hence, even a single update step can quickly improve the model performance on a large quantity of a user’s data, thus facilitating rapid personalization” is not well understandable. Please provide explanation in more detail.